# Caregiver Burden among Family Caregivers of Cancer Survivors Aged 75 Years or Older in Japan: A Pilot Study

**DOI:** 10.3390/healthcare11040473

**Published:** 2023-02-06

**Authors:** Yoshiko Kitamura, Hisao Nakai, Yukie Maekawa, Hisako Yonezawa, Kazuko Kitamura, Tomoe Hashimoto, Yoshiharu Motoo

**Affiliations:** 1School of Nursing, Kanazawa Medical University, Kahoku 920-0293, Japan; 2Kanazawa Medical University Hospital, Kahoku 920-0293, Japan; 3Komatsu Sophia Hospital, Komatsu 923-0861, Japan

**Keywords:** family caregivers, caregiver burden, cancer survivors, aged, Japan

## Abstract

The purpose of this study was to assess the burden of caregiving among family caregivers of cancer survivors aged 75 years or older in Japan. We included family caregivers of cancer survivors aged 75 years or older who were attending two hospitals in Ishikawa Prefecture, Japan, or receiving treatment during home visits. A self-administered questionnaire was developed based on previous studies. We obtained 37 responses from 37 respondents. Excluding those with incomplete responses, we had data from 35 respondents for analysis. The factor that significantly influenced the burden of caregiving for cancer survivors aged 75 years or older and family caregivers living together was the provision of full-time care (*p* = 0.041). Helping cancer survivors manage money (*p* = 0.055) was also associated with a higher burden. For family caregivers living separately, a more detailed examination of the association between the sense of caregiving burden and distance of travel to provide home-visit care is necessary, along with more support to attend hospitals with cancer survivors.

## 1. Introduction

The number of people aged 65 and over is increasing worldwide. The average life expectancy in the world has increased by more than 8 years since 1990, reaching 72.6 years in 2019 and is expected to reach 77.1 years by 2050 [1]. When older people become frail, families are often the first to provide care, and many of these family members are likely to be working [2]. Family caregivers often provide informal and unpaid care [3,4,5,6], frequently living with the care recipient, and spending a lot of time caring for them [3,4]. However, they may not be able to provide all the care needed. They may also experience both a physical and psychological burden that may affect their health. Those caring for a family member with an irreversible and progressive illness, particularly dementia, may experience a deterioration in their health, including a nervous breakdown and sleep disturbances [7,8]. Caring for a family member with Alzheimer’s disease can cause stress, anxiety, and depression in family caregivers [9]. Those who care for older people with mental health problems, provide long-term care, and have little social support are at increased risk of mental health problems themselves, including depression [10,11]. A study in a Thai rural community found that the predictors of caregiver burden were the care recipient’s ability to perform activities of daily living, the depression score of the caregiver, and the total hours of care provided [12]. The employment status of family caregivers has also been cited as a predictor of the care burden of older adults [13]. The burden of caregiving for working family caregivers includes both the direct burden of caregiving, and changes in their roles and employment, as well as schedule disruptions due to frequent visits to hospitals and clinics [14]. The caregiver burden is therefore affected by whether family caregivers are working.

The number of cancer survivors continues to increase because of advances in early detection and treatment and the aging and growth of the population [15]. Estimates to the year 2035 indicate that the number of older cancer survivors may increase worldwide. The largest relative increases in incidence are predicted in the Middle East and Northern Africa (+157%), and China (+155%) [16]. Older cancer survivors often have underlying medical conditions in addition to cancer and require complex healthcare provisions. This places a high demand on their caregivers [17]. The burden on family caregivers may also be affected by the increasing immobility of cancer patients if their condition deteriorates [18]. Family caregivers who spend a lot of time with cancer patients have been shown to experience psychological and physical health problems due to the strain and burden of caregiving [19]. Caregivers caring for a cancer patient while raising children, and working family caregivers may also experience their own physical and psychological health issues [20].

The aging of the population in Japan is progressing at a rate unparalleled in other countries. The proportion of people over 65 years old in Japan is 28.8% [21], and cancer is the most common cause of death among all Japanese and those over 75 years old [22]. The number of older people in Japan is expected to continue to increase until 2036 [21]. The life expectancy of cancer survivors is also improving [23]. Caring for cancer survivors is therefore a serious issue. The number of older people living alone or in married-couple households is also increasing in Japan [24], and the number of adult children living apart from their parents but still providing care is increasing.

The purpose of this study was to understand the burden of caregiving among family caregivers of cancer survivors aged 75 years or older in Japan. This study was also a pilot study for a future survey. We will determine the survey questions from this study. The survey will then aim to clarify the characteristics of family caregivers and their sense of burden in caring for cancer survivors aged 75 years or older who are living at home in Ishikawa Prefecture, Japan.

## 2. Materials and Methods

### 2.1. Study Design and Participants

This was a cross-sectional study using self-administered questionnaires. We included family caregivers of cancer survivors aged 75 years or older who were attending two hospitals that provide care for cancer survivors in Ishikawa Prefecture, Japan, or receiving treatment during home visits. Family caregivers were invited to participate by physicians from the two hospitals. Ishikawa Prefecture is in the center of the Hokuriku region facing the Sea of Japan, with the Noto peninsula jutting out into the Sea of Japan to the north [25]. The population of Ishikawa Prefecture is approximately 1.12 million, and around 30% of them are over 65 years old [26].

### 2.2. Data Collection

For this study, we developed a self-administered questionnaire based on previous studies. We used the Lawton Instrumental Activities of Daily Living [27] as a reference to investigate the care provided by family caregivers with activities of daily living of cancer survivors aged 75 years or older. The Burden Index of Caregiver-11 (BIC-11) was used to measure the sense of caregiving burden. The BIC-11 is a multidimensional scale that measures the sense of burden among caregivers who care for someone at home. The BIC-11 was created as a unique Japanese caregiver burden scale. The scale consists of five domains: time-dependent burden, emotional burden, existential burden, physical burden, and service-related burden. Each domain consists of two questions and 10 sub-items. This gives a total of 11 items, including the total care burden [28]. The total score ranges from 0 to 44, with higher scores indicating a greater burden on family caregivers [29]. The validity and reliability of the BIC-11 have been confirmed [28]. This study was conducted from 1 March to 31 March 2022.

### 2.3. Survey Details

#### 2.3.1. Family Caregivers’ Background

The basic attributes taken about family caregivers were age and sex. We also asked them whether they were living with the cancer survivor to whom they provided care using the options: “living together” or “living separately”. Their options for employment type were “full-time”, “part-time”, “unemployed”, or “other”. Annual income was classified into three categories based on the distribution of the annual income of older people’s households in Japan: “less than 3.18 million yen”, “between 3.18 million yen and 3.48 million yen”, and “3.48 million yen or more” [30]. Health status was categorized as “good”, “fairly good”, “somewhat poor” and “poor”. Respondents were asked to indicate whether they had any chronic conditions using three options: “yes”, “no” and “don’t know”.

#### 2.3.2. Background of Cancer Survivors Aged 75 Years and Older

The caregivers were asked to provide information about the basic attributes of their care recipient, such as their age, sex, and relationship with the family caregiver. Respondents selected treatment history by treatment method from “surgery”, “radiation therapy”, “chemotherapy”, and “other”. They were also asked whether the care recipient had any diseases other than cancer using, responding “yes”, “no”, or “don’t know”. We also asked if the care recipient had a diagnosis of dementia (possible responses were “yes”, “no”, and “don’t know”).

#### 2.3.3. Family Caregiver Care Status

The respondents were asked how long they had been taking care of family members; there were four response categories: “less than 1 year”, “1–3 years”, “3–5 years”, and “5 years or more”. Respondents were asked about the number of times they had to get up at night to provide care in the past month; there were four categories: “often”, “sometimes”, “almost never”, and “never”. The respondents were asked if they had experienced difficulties doing other household chores and jobs because of caregiving in the past month; there were four categories: “often”, “sometimes”, “almost never”, and “never”. Respondents were asked about care partners and care advisors, both with responses of “present” or “absent”.

#### 2.3.4. Family Caregiver Care Details

The respondents were asked to answer “yes” or “no” to say if they provided help with “making phone calls”, “shopping”, “meal preparation”, “eating meals”, “cleaning”, “dressing”, “bathing”, “using the toilet”, “defecation (including handling enemas and suppositories)”, “urination (including handling the urinal)”, “changing clothes”, “laundry”, “transportation to and from to hospital”, “walking (including accompanying and operating wheelchair)”, “getting in or out of bed”, “medication management”, “money management”, and “advising about concerns”.

#### 2.3.5. Family Caregivers’ Sense of Caregiving Burden

The BIC-11 was used to assess family caregivers’ sense of caregiving burden. Items included: “I don’t have enough time for myself because of caregiving”, “I can’t go out freely because of caregiving”, “I get tired of everything when I am a caregiver”, “I want to leave caregiving to someone else”, “It is hard because I don’t feel fulfilled when I am a caregiver”, “It is hard because I don’t find meaning in caring for my family member”, “I feel physical pain when providing care”, “My health has suffered because of caregiving”, “I don’t feel like caring for my family member”, and “I feel like I want to leave the work to someone else”. Other items include “I am troubled because patients do not want caregiving services”, “It is a burden that caregiving services come into my house” and “Overall, how much of a burden do you think caregiving is on you?”. All responses used a five-point Likert-type scale (0 = never, 1 = almost never, 2 = sometimes, 3 = often, 4 = always).

#### 2.3.6. Support Required by Family Caregivers

The respondents were asked to comment freely on the support they required.

### 2.4. Analysis Methods

After obtaining the distribution of the background of family caregivers and cancer survivors aged 75 years or older, we defined the employment type of family caregivers as “full-time” for those who answered “full-time” and “other” for all other responses. Annual income was defined as “less than 3.18 million yen” for “less than 3.18 million yen” and “other” for all other responses. Health status was classified into two categories: general health status into “good” for “good/fairly good” and “other” for “somewhat poor/poor”, and chronic conditions into “yes” for those who responded “yes”, and “other” for responses of “no” or “don’t know”. The background of cancer survivors aged 75 years or older were analyzed by classifying a diagnosis of dementia into “yes” and “other” (for responses of “no/don’t know”). The duration of care was “less than 1 year” and “other” (for “1–3 years”, “3–5 years”, and “5 years or more”). Respondents who answered “often/sometimes” to the number of times they had to get up at night to provide care in the past month were grouped into “yes”, and those who answered “almost never/never” into “other”. BIC-11 uses a five-point Likert-type scale (0 = never, 1 = almost never, 2 = sometimes, 3 = often, 4 = always). After obtaining the distribution of the BIC-11 scores, we divided the group into two using the median value, giving a no or low care burden group and a high care burden group, in line with a previous study [31].

Overall, 32 (91.4%) of the study participants lived with a cancer survivor aged 75 years or older. Three family caregivers (8.6%) who lived separately were excluded to control for the effect of residential status on the burden of caregiving. We used the chi-square test or Fisher’s direct probability test to examine the association between the other items and the sense of caregiving burden of family caregivers living with the care recipient as an objective variable. The significance level was set at 5%. We used SPSS Ver. 27 (IBM Corp., Armonk, NY, USA) for statistical analysis. The support currently required by family caregivers was categorized by the type of residence.

### 2.5. Ethical Considerations

This study was carried out with the approval of the University Medical Research Ethics Review Committees at the authors’ universities (No. I692). The participants were given a written informed consent form and were informed of the purpose and importance of the study, the survey method, the fact that participation was voluntary, and the fact that they would not be personally identified when the results were made public. Participants completed a self-administered questionnaire. Completion of the questionnaire implied their consent.

## 3. Results

Overall, 60 family caregivers were asked to participate in the survey and 37 responded (response rate: 61.7%), with 35 respondents (94.6%) answering all the items. The mean age (standard deviation) of the family caregivers was 68.9 (11.1) years, four were men (11.4%), and 31 were women (88.6%). For living arrangements, 32 (91.4%) were living with their care recipient and three (8.6%) separately. The mean (standard deviation) age of cancer survivors over 75 years was 79.9 (4.1) years, 27 (77.1%) were men and eight (22.9%) were women. The backgrounds of family caregivers and cancer survivors are shown in Table 1.

### 3.1. Factors Associated with BIC-11 Score of Family Caregivers Living with the Care Recipient (n = 32)

Overall, 32 family caregivers were living with the cancer survivor, and their mean age (standard deviation) was 70.4 (10.0) years. They included three men (8.4%) and 29 women (90.6%). The median (range) of BIC-11 was 2.0 (0–28). The distribution of BIC-11 was 0 = 12 (37.5%), 1 = 3 (9.4%), 2 = 3 (9.4%), and ≥ 3 = 14 (43.7%). The results of the univariate analysis are shown in Table 2. Eight (25.0%) full-time employees (*p* = 0.041) had a significantly higher percentage of high BIC-11, as did the 12 respondents (37.5%) who provided money management help for cancer survivors (*p* = 0.055). Table 2 shows the results of the cross-tabulation.

### 3.2. Background of Family Caregivers Living Separately from the Care Recipient (n = 3)

The mean age (standard deviation) was 50.7 years (3.2). There was one man (33.3%) and two women (66.7%).

### 3.3. Support Required by Family Caregivers (Free Answer) (n = 3)

The family caregivers who lived with their fathers indicated that they needed information about available caregiver support as soon as possible, to be listened to, and to have support with helping the caregiver to take baths and for housework. The family caregivers who lived apart from their cancer survivors mentioned that they needed help to reduce the burden of taking their care recipient to and from the doctor’s office once a week, which took 3 h each way, transportation expenses for visiting the doctor, and support for accompanying the care recipients when they visit the doctor.

## 4. Discussion

Our study aimed to understand the burden of caregiving among family caregivers of cancer survivors aged 75 years or older who receive home care, hospital visits, or home visits in Ishikawa Prefecture, Japan. The mean age of the participants in the study by Sugiyama et al. on family caregivers with Japanese cancer survivors was 48 years [32]. The mean age of the participants in this study was 68.9 years, which may have been influenced by the fact that this study was conducted among family caregivers of cancer survivors aged 75 years or more.

Many studies have reported the relationship between employment and family caregivers’ sense of caregiving burden [33,34,35,36]. In particular, it has been pointed out that the physical functions of cancer patients decline at the end of life, which makes family caregivers more anxious, increases their sense of caregiving burden, and has a negative effect on their employment [37,38]. In our study, working full-time was associated with a high care burden, but the direction of the relationship is unclear. However, full-time work may be an important factor when considering the burden of providing care.

When older adults rely on their children for financial support and caregiving, their children’s physical and mental health is threatened and family relationships are negatively affected [39,40]. We found that providing money management support was associated with an increased burden among family caregivers. It is not clear why this should be the case, and this will need further investigation in future studies.

One free text comment from family caregivers who lived apart from their care recipient noted that the 3-h each-way trip by private car and long outpatient visits were a burden. In a previous study, the average distance traveled by cancer survivors aged 75 years or older to receive outpatient chemotherapy in Ishikawa Prefecture, Japan, was 40.7 km [41]. In Japan, the physical burden and fatigue of patients who travel long distances to receive outpatient chemotherapy are issues [42]. Our findings suggest that long-distance travel may also be a burden for family caregivers who live separately. Only three family caregivers were living apart from their cancer survivors, so the relationship with caregiver burden is unknown at this time, but those who support cancer survivors need to be aware of the burden on caregivers of providing care and attending long outpatient visits with the patient.

Previous results suggest that family caregivers are also older and at risk of developing cancer themselves [43]. Another study found that the improved life expectancy of cancer survivors [23] means it is necessary to clarify the burden of family caregivers who work full-time, manage the money of the cancer survivor, and travel to the cancer survivor’s home from neighboring cities to provide care. Our study supports these findings.

This study had several limitations. First, the total number of respondents was small. Only three family caregivers lived separately from their care recipients. Second, most of the results have a gender bias, because the majority of caregivers were female (88.6%). Third, the majority of study participants were family caregivers living with their cancer survivors and with a low caregiving burden. Many of the cancer survivors over 75 years of age living with their caregivers may have had a good ability to perform activities of daily living and therefore presented a low physical caregiving burden. Fourth, the BIC-11 is a scale suitable for measuring the burden of caregivers who care for a family member at home [28]. It may not be suitable for measuring the burden of caregiving among family caregivers of cancer survivors. Fifth, information on cancer survivors was reported by the family caregivers, and not the cancer survivors themselves, which may have introduced some bias or inaccuracy. Sixth, this was a cross-sectional study, and it is therefore not possible to establish any causal relationships between the study variables.

## 5. Conclusions

Our findings indicate that working full-time and helping cancer survivors with money management may be associated with a care burden. The distance traveled by family caregivers to provide care may also be a factor. In the future, it will be necessary to investigate the sense of caregiving burden by considering patterns of work and money management support for cancer survivors among family caregivers who live with their cancer survivors. Additionally, the number of participants should be increased to include younger caregivers and urban and rural caregivers. The method of measuring the sense of caregiving burden also needs to be re-examined. The number of older cancer survivors living alone or in married-couple households is increasing in Japan. The relationship between the distance traveled by family caregivers to provide care and support, especially with hospital visits, and their sense of caregiver burden should therefore be investigated.

## Figures and Tables

**Table 1 healthcare-11-00473-t001:** Background of family caregivers and cancer survivors over 75 years (n = 35).

	Item		*n*	(%)
Family caregiver background				
	Age (median [range]), years	74.0 (47–82)		
	Sex	Men	4	(11.4)
		Women	31	(88.6)
	Living arrangements with cancer survivors aged 75 years and older	Living together	32	(91.4)
		Living separately	3	(8.6)
Background of cancer survivors aged 75 years and older				
	Age (median [range]), years	79.0 (75–95)		
	Sex	Men	27	(77.1)
		Women	8	(22.9)
	Relationship with family caregiver			
		Husband	23	(65.7)
		Mother	8	(22.9)
		Father	3	(8.6)
		Father-in-law	1	(2.9)
	Treatment history by treatment method (multiple answers allowed)			
		Chemotherapy	27	(77.1)
		Surgery	16	(45.7)
		Radiation therapy	11	(31.4)
		Other	1	(2.9)
	Diseases other than cancer	Yes Other	13 22	(37.1) (62.9)

**Table 2 healthcare-11-00473-t002:** Background and caregiving status of family caregivers and cancer survivors in relation to BIC-11 (n = 32).

					Burden of Care (BIC-11)				
	Item	Category	Total		No or Low Group		High Group		
			*n*	(%)	*n*	(%)	*n*	(%)	*p* Value
Family caregiver’s basic attributes, work status, annual income, health status, pre-existing conditions									
	Age	Average	32	(100.0)	15	(46.9)	17	(53.1)	0.389 ^1^
	Sex	Men	3	(9.4)	1	(33.3)	2	(66.7)	1.000 ^2^
		Women	29	(90.6)	14	(48.3)	15	(51.7)	
	Employment type	Full-time	8	(25.0)	1	(12.5)	7	(87.5)	0.041 ^2^
		Other	24	(75.0)	14	(58.3)	10	(41.7)	
	Annual income	Less than 3.18 million yen	24	(75.0)	13	(54.2)	11	(45.8)	0.229 ^2^
		Other	8	(25.0)	2	(25.0)	6	(75.0)	
	Status of health	Good	24	(75.0)	11	(45.8)	13	(54.2)	1.000 ^2^
		Other	8	(25.0)	4	(50.0)	4	(50.0)	
	Chronic conditions	Yes	12	(37.5)	6	(50.0)	6	(50.0)	0.784 ^1^
		Other	20	(62.5)	9	(45.0)	11	(55.0)	
Attributes of cancer survivors aged 75 years or older									
	Age	Average	32	(100.0)	15	(46.9)	17	(53.1)	0.433 ^1^
	Sex	Men	25	(78.1)	11	(44.0)	14	(56.0)	0.424 ^2^
		Women	7	(21.9)	4	(57.1)	3	(42.9)	
	Diagnosis of dementia	Yes	3	(9.4)	1	(33.3)	2	(66.7)	1.000 ^2^
		Other	29	(90.6)	14	(48.3)	15	(51.7)	
Family caregiver status									
	Period providing care	Less than 1 year	16	(50.0)	8	(50.0)	8	(50.0)	0.723 ^1^
		Other	16	(50.0)	7	(43.8)	9	(56.3)	
	In the past month, have you had to get up at night to provide care?	Yes	6	(18.8)	1	(16.7)	5	(83.3)	0.178 ^2^
		Other	26	(81.3)	14	(53.8)	12	(46.2)	
	In the past month, has caregiving made it difficult for you to do other household chores or jobs?	Yes	9	(28.1)	6	(66.7)	3	(33.3)	0.243 ^2^
		Other	23	(71.9)	9	(39.1)	14	(60.9)	
	Care partners	Present	24	(75.0)	12	(50.0)	12	(50.0)	0.691 ^2^
		Absent	8	(25.0)	3	(37.5)	5	(62.5)	
	Care advisors	Present	27	(84.4)	14	(51.9)	13	(48.1)	0.338 ^2^
		Absent	5	(15.6)	1	(20.0)	4	(80.0)	
Care provided by family caregiver									
	Making phone calls	Yes	9	(28.1)	3	(33.3)	6	(66.7)	0.444 ^2^
		No	23	(71.9)	12	(52.2)	11	(47.8)	
	Shopping	Yes	10	(31.3)	5	(50.0)	5	(50.0)	1.000 ^2^
		No	22	(68.8)	10	(45.5)	12	(54.5)	
	Meal preparation	Yes	22	(68.8)	9	(40.9)	13	(59.1)	0.450 ^2^
		No	10	(31.3)	6	(60.0)	4	(40.0)	
	Eating meals	Yes	4	(12.5)	2	(50.0)	2	(50.0)	1.000 ^2^
		No	28	(87.5)	13	(46.4)	15	(53.6)	
	Cleaning	Yes	16	(50.0)	6	(37.5)	10	(62.5)	0.288 ^1^
		No	16	(50.0)	9	(56.3)	7	(43.8)	
	Dressing	Yes	3	(9.4)	1	(33.3)	2	(66.7)	1.000 ^2^
		No	29	(90.6)	14	(48.3)	15	(51.7)	
	Bathing	Yes	5	(15.6)	4	(80.0)	1	(20.0)	0.161 ^2^
		No	27	(84.4)	11	(40.7)	16	(59.3)	
	Using the toilet	Yes	5	(15.6)	2	(40.0)	3	(60.0)	1.000 ^2^
		No	27	(84.4)	13	(48.1)	14	(51.9)	
	Defecation (including handling enemas and suppositories)	Yes	2	(6.3)	1	(50.0)	1	(50.0)	1.000 ^2^
		No	30	(93.8)	14	(46.7)	16	(53.3)	
	Urination (including handling the urinal)	Yes	3	(9.4)	1	(33.3)	2	(66.7)	1.000 ^2^
		No	29	(90.6)	14	(48.3)	15	(51.7)	
	Changing clothes	Yes	7	(21.9)	3	(42.9)	4	(57.1)	1.000 ^2^
		No	25	(78.1)	12	(48.0)	13	(52.0)	
	Laundry	Yes	19	(59.4)	9	(47.4)	10	(52.6)	0.946 ^1^
		No	13	(40.6)	6	(46.2)	7	(53.8)	
	Transportation to and from the hospital	Yes	12	(37.5)	6	(50.0)	6	(50.0)	0.784 ^1^
		No	20	(62.5)	9	(45.0)	11	(55.0)	
	Walking (including accompanying and operating wheelchair)	Yes	2	(6.3)	2	(100.0)	0	(0.0)	0.212 ^2^
		No	30	(93.8)	13	(43.3)	17	(56.7)	
	Getting in or out of bed	Yes	3	(9.4)	1	(33.3)	2	(66.7)	1.000 ^2^
		No	29	(90.6)	14	(48.3)	15	(51.7)	
	Medication management	Yes	12	(37.5)	5	(41.7)	7	(58.3)	0.647 ^1^
		No	20	(62.5)	10	(50.0)	10	(50.0)	
	Money management	Yes	12	(37.5)	3	(25.0)	9	(75.0)	0.055 ^1^
		No	20	(62.5)	12	(60.0)	8	(40.0)	
	Advising about concerns	Yes	7	(21.9)	3	(42.9)	4	(57.1)	1.000 ^2^
		No	25	(78.1)	12	(48.0)	13	(52.0)	

^1^ χ^2^ test, ^2^ Fisher’s exact test.

## Data Availability

The data analyzed during this study are included in this published article. Further inquiries can be directed to the corresponding authors.

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
