# Peer review of "Caregiver Burden among Family Caregivers of Cancer Survivors Aged 75 Years or Older in Japan: A Pilot Study"

_healthcare, 2023, doi:10.3390/healthcare11040473_

Round 1

Reviewer 1 Report

The article deals with a very acrual topic. In modern societies, the number of elderly people is increasing, including cancer survivors. It is therefore reasonable to examine and describe the burdens that people who care for the sick bear. The article is in the form of a pilot study, and the authors outline further research problems at the end.

The study group was selected correctly. The inclusion and exclusion criteria have been clearly formulated. 37 people surveyed constitute a sufficient group for the obtained data to be reliable and to be statistically processed. Of course, an increase in the number of people surveyed would affect the quality of the submitted study, which the authors are aware of. The questionnaire was developed correctly. It included questions about the demographic data and financial status of the patient and caregiver, diagnosis and methods of therapy, and the scope of support provided to the sick person.

Noteworthy is the respect for ethical aspects. The article contains a reference to legal acts and conventions as well as information about the consent of the ethics committee. Its location and consent number are given, which should be standard, and is not always the case in the publications I have reviewed so far.

Reviewer 2 Report

Thank you for the opportunity to review your manuscript titled Caregiver burden among family caregivers of cancer survivors aged 75 years or older in Japan: a pilot study. It's a 'pilot study' to try to clarify the characteristics of family caregivers and their sense of 63 burden in caring for cancer survivors aged 75 years or older who are living at home in 64 Ishikawa Prefecture, Japan.

Please use keywords as MeSH terms. One MeSH term would be Caregiver burden.

Methods section. How was the sample selected?

Discussion. Limitations. It should be noted that most of the results have a gender bias, as the majority are female caregivers (88.6%).

Conclusions. Do not start the conclusions by indicating what should be done in the future. Respond to the stated objective and, finally, indicate what future proposals you propose to investigate.

Reviewer 3 Report

This study should be published because the theme is critical for healthcare systems/providers and social workers and organizations. The ageing population is an issue for caregiving because human ageing is also associated with frailty. Cancer disease is an issue for older people and their caregiving, especially for family caregivers. Although the article is about cancer survivors, the cancer experience is always a physical and psychological burden both for patients and their relatives. Furthermore, there are many reasons to determine those burdens: knowledge and literacy about cancer caregiving, financial support to provide formal caregiving and healthcare, emotional ties to the relative (story of life), house conditions, social network (neighbourhoods, family, friends), etc. I would suggest the following reviews to increase the article’s quality and impact:

Introduction

1.       I would suggest changing the sentence “Older people often experience declining health, and this places a heavy burden on their families” because the sentence has a negative sense of ageing and older people. Neither are old people constantly sick nor are families always carers. My suggestion: “Facing the frailty of older people, families are often the first caregiving providers…

2.       I would suggest finding recent bibliography about family caregivers and burdens related to older people (Lines 27 to 40).

a.       Riffin C, Van Ness PH, Wolff JL, Fried T. Multifactorial Examination of Caregiver Burden in a National Sample of Family and Unpaid Caregivers. J Am Geriatr Soc. 2019 Feb;67(2):277-283. doi: 10.1111/jgs.15664. Epub 2018 Nov 19. PMID: 30452088; PMCID: PMC6367031.

b.       Arruda MS, Macedo MNGF, Ottaviani AC, Nunes DP, Cardoso JFZ, Santos KCD, Brito TRP, Santos-Orlandi AAD. Correlation of family functionality and burden of informal caregivers of hospitalized older adults. Rev Gaucha Enferm. 2022 Jul 31;43:e20210081. English, Portuguese. doi: 10.1590/1983-1447.2022.20210081.en. PMID: 35920478.

c.       Tuttle D, Griffiths J, Kaunnil A. Predictors of caregiver burden in caregivers of older people with physical disabilities in a rural community. PLoS One. 2022 Nov 4;17(11):e0277177. doi: 10.1371/journal.pone.0277177. PMID: 36331949; PMCID: PMC9635719.

d.       Khalaila R. Caregiver Burden and Compassion Fatigue Among Arab Family Caregivers of Older Relatives. J Appl Gerontol. 2021 Jul;40(7):722-730. doi: 10.1177/0733464820920100. Epub 2020 May 29. PMID: 32468896.

e.       Manzini CSS, do Vale FAC. Emotional disorders evidenced by family caregivers of older people with Alzheimer's disease. Dement Neuropsychol. 2020 Jan-Mar;14(1):56-61. doi: 10.1590/1980-57642020dn14-010009. PMID: 32206199; PMCID: PMC7077868.

f.        So MKP, Yuk H, Tiwari A, Cheung STY, Chu AMY. Predicting the burden of family caregivers from their individual characteristics. Inform Health Soc Care. 2022 Apr 3;47(2):211-222. doi: 10.1080/17538157.2021.1988955. Epub 2021 Oct 28. PMID: 34709118.

3.       I would suggest removing the reference to “especially working-age people” (L28) and “working family caregivers” (L38) because you didn’t analyze them. Indeed, if family caregiving is a burden, this burden could increase if relatives are employees. I suggest having one sentence that refers to it.

4.       I would suggest explaining better and deeply the idea: “the number of older cancer survivors may increase worldwide” (L43). What this is meaning? Do Older people have more life expectancy after cancer than other generations?

Materials and Methods

1.       I would suggest another organization: 2.1) Type of study (descriptive, experimental, analytic, ??); 2.2) Data Collection (including survey details); 2.3) Recruitment process (how you recruit the participants?); 2.4. Ethical considerations.

Results & Discussion

1.       I would suggest reviewing the reference to “rural areas” in the discussion (L226). You never did a reference before. Your survey didn’t ask about location, so it’s not a variable of your study. I would suggest transferring the first paragraph to the conclusion, namely lines 222 to 228.

a.       I would suggest a conclusion like: “The study should increase the number of participants to include young caregivers (like the Sugiyama et al. study) and caregivers from urban and rural areas”.

2.        The same suggestion for your reference “employment” (L229), “full-time work” (L233)

a.       During your study, there is no differentiation between caregivers who have a job and others who don’t have one. So, there are no data to discuss it.

b.       When you refer “work full-time (L249), are you refer “caregiving” or a “job”?

3.       Clarify the references “financial support” and “money management”. The BCI-11 is about supporting financial management (e.g., paying monthly accounts with the care recipients’ money) and not “financial support” (that means giving money to the care recipients because he/she need it).

4.       I would suggest a discussion that clarifies the association between the open question and the BIC-11 answers. 
